# Beyond Assuming Co-Benefits in Nature-Based Solutions: A Human-Centered Approach to Optimize Social and Ecological Outcomes for Advancing Sustainable Urban Planning

**Agathe Colléony *** and **Assaf Shwartz**

Human and Biodiversity Research Lab, Faculty of Architecture and Town Planning, Technion–Israel Institute of Technology, Haifa 32000, Israel

*   Correspondence: agathe.colleony@gmail.com

**Abstract:** Urbanization deletes and degrades natural ecosystems, threatens biodiversity, and alienates people from the experience of nature. Nature-based solutions (NbS) that are inspired and supported by nature have the potential to deliver multifunctional environmental and social benefits to address these challenges in urban areas under context-specific conditions. NbS implementation often relies on a one-size-fits-all approach, although interventions that maximize one benefit (e.g., biodiversity conservation) may have no influence on, or even negatively affect, others (e.g., social justice). Furthermore, the current pathways from NbS to various benefits do not rely on a deep understanding of the underlying processes, prohibiting the identification of optimal solutions that maximize synergies across pathways. We present a comprehensive socio-ecological framework that addresses these issues by recognizing that cities are human-dominated environments that are foremost built and maintained to support humans. Our framework demonstrates how we can use experiments and niche species models to understand and predict where species will be and where people will be healthy and happy in a comparable manner. This knowledge can then be integrated into decision support tools that use optimization algorithms to understand trade-offs, identify synergies, and provide planners with the tools needed to tailor context-specific NbS to yield greener, more resilient cities with happier people and reduced inequality.

**Keywords:** green city; ecosystem services; resilient city; ecological indicators; specific components of nature; human well-being; systematic conservation planning

## 1. Introduction

Urbanization is emerging as a major contemporary issue worldwide, with strong implications on the health of humans and natural ecosystems [1]. One promising way to address this challenge is to green cities by adopting Nature-based Solutions (NbS) in the design and management of new and existing urban areas. NbS can be defined as "solutions that are inspired and supported by nature, which are cost-effective, simultaneously provide environmental, social and economic benefits and help build resilience" [2]. They bring nature back into cities by promoting the creation, maintenance, enhancement, and restoration of ecosystems, supporting biodiversity conservation and providing multiple ecosystem services to citizens (e.g., climate mitigation, water management, air quality health, and well-being [3]). For instance, planting trees along streets and roads can contribute to carbon sequestration, reduce the urban heat island effect as well as noise and air pollution, provide a habitat for birds and bats that, in turn, regulates the pest population (e.g., mosquitos), and promote public health through psychological restoration (e.g., [4–6]). The growing recognition of the potential for nature

to address multiple concerns has led many cities across the world to implement NbS [7]. However, despite rising interest and deployment of NbS, there is still a striking lack of empirical results on the cost-effectiveness of NbS, particularly with regard to their ability to generate co-benefits and their impacts on health and well-being [8,9]. This knowledge is crucial for the design and implementation of NbS that are socially comprehensible and acceptable to many stakeholders [3].

The numerous benefits of greening cities, through NbS, can be classified into three pathways that mirror the main motivations often raised by scientists, policy makers, planners, and the general public for integrating nature in cities (Figure 1) [10–12]. The first pathway is focused on directly enhancing biodiversity by restoring habitats and connectivity through green infrastructures, thus reducing the ecologically detrimental impacts of urbanization at the local scale. A second pathway is linked to ecosystem services and sustainable development frameworks. Green infrastructures can be designed to provide several services that can help to mitigate some local and global environmental challenges caused by urban development and lifestyle. The third pathway focuses on biophilic designs that provide people with more meaningful experiences of nature, aiming to enhance both people's health and well-being derived from their interactions with nature and their affinity with nature and its protection (an indirect ecological benefit). For instance, green roofs can restore grassland habitats (pathway 1), reduce runoff (pathway 2), and be designed for nature-based recreation (pathway 3). These three pathways are not entirely distinct, and some overlaps, synergies, and trade-offs can be found between the outcomes of the three pathways (e.g., the enhancement of biodiversity for ecological benefits has the potential to increase well-being benefits, but this relationship may then decline for areas with a high density of natural elements).

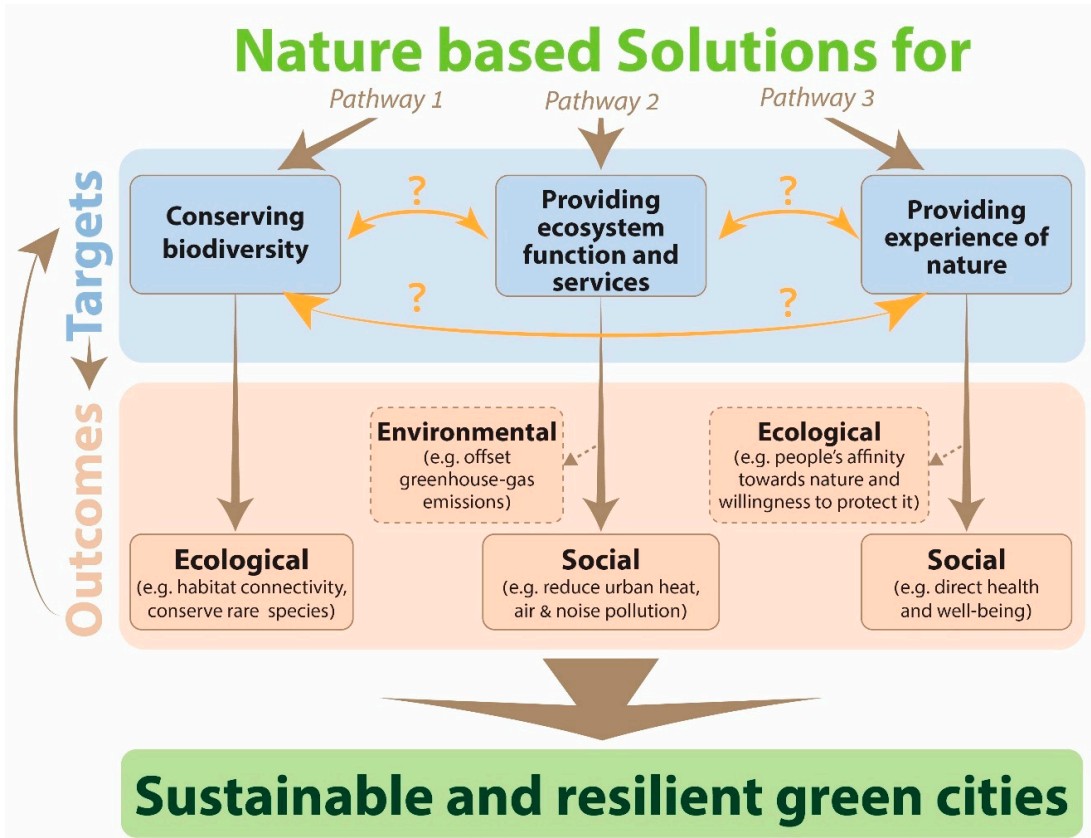

**Figure 1.** A schematic flowchart describing the three suggested pathways for greening cities and the achievement of sustainability. The brown arrows demonstrate the benefits of each pathway and the dashed arrows represent indirect benefits. The orange arrows refer to trade-offs between pathways, and the question marks illustrate the knowledge gaps in the understanding and integration of those trade-offs.

Evidence for overlaps and synergies between the different pathways (e.g., [13,14]) has led to the development of the NbS concept that assumes simultaneous delivery of social and environmental benefits. In fact, urban ecologists, policy makers, and planners often advocate that targeting one pathway (i.e., biodiversity conservation) can maximize others as well [10]. This argument relies on the assumption that "green infrastructure" equals "nature" which equals "biodiversity" which equals "ecosystem services" [15,16]. However, nature is not a panacea and, to date, evidence has demonstrated that this assumption is not necessarily true [17], and various knowledge gaps and trade-offs both between and within pathways occur. Particularly, we still have very limited understanding of the mechanisms that drive the relationship between NbS and multiple components of health and well-being [18]. As a result, planners often adopt a one-size-fits-all approach by setting up broad targets, assuming co-benefits, and ignoring spatial context and moderating variables [19]. Failing to reach those targets could seriously undermine the whole concept of NbS and the design of sustainable green cities.

In search of a solution, Raymond et al. [3] recently highlighted important gaps in existing frameworks and suggested what needs to be done, as follows: (1) assess the impacts of NbS "within and across different societal and environmental challenges"; (2) develop and integrate novel tools to map and reliably upscale and predict the spatial distributions of different NbS outcomes, based on land-use scenarios and taking into account the socio-economic context; and (3) develop a "decision making toolkit that simplifies and systematizes the monitoring and evaluation of co-benefits in decision support". The aim of this essay is to propose *how* this can be achieved. We propose a novel framework that suggests a paradigm shift from assuming co-benefits or seeking to directly relate two or more outcomes, to a more holistic approach that uses trade-offs and synergies to inform policy and decision-making. The only way to achieve this is by integrating theories and methods from several disciplines to direct future interdisciplinary research to shed light on the mechanisms that drive the relationships between NbS and several outcomes. Our framework will also demonstrate how planners can move away from a one-size-fits-all approach to optimize NbS in a given context to yield greener, more resilient cities with happier people and reduced inequality. In this essay, we first provide a synthesis of the literature on the synergies and trade-offs within and between the pathways. We then present our novel framework and highlight how it can help to advance both research and practice to strengthen the implementation of NbS that maximize co-benefits.

## 2. Synthesis of the Literature: Pathways, Synergies, and Trade-offs

### 2.1. Complexities within the Three Pathways for Greening Cities

Ecologists have identified interventions that can locally maintain or even increase urban biodiversity, provide habitats for animals and plants, and restore connectivity within the urban matrix [20]. This could, in turn, reduce the detrimental ecological impacts of urbanization (pathway 1, Figure 1). However, different taxonomic groups may respond differently to the implementation of NbS. For instance, in public green spaces in Paris, France, bird richness was positively associated with tree cover, while both native plants and butterfly richness were found to decrease as tree cover increased [21]. Furthermore, harboring a rich biodiversity or even endangered species in cities is not a guarantee of effective conservation. Urban populations are not necessarily viable, and some NbS may create ecological traps, i.e., scenarios in which animals mistakenly prefer a habitat where their fitness is lower than in other available habitats following rapid environmental change [5,22]. For example, the setting of bird nesting boxes can locally increase nesting rates of a species in a habitat that is not suitable for it [23]. Finally, it is not yet clear whether NbS that enhance biodiversity locally (e.g., by increasing the green index of cities) can reduce the impacts on biodiversity at the landscape or regional spatial scales. Compact urban development will reduce the spatial extent of developed areas, while under a lower urbanization intensity, the ecological impacts are reduced locally but spread over larger

areas [24,25]. This questions the extent to which NbS that result in greener and more biodiverse cities can contribute to general conservation efforts if they generate more land degradation.

To date, much of the NbS literature has focused on the idea that integrating nature into cities can provide provisioning and regulating ecosystem services to mitigate some key societal challenges [3]. For instance, NbS can help to offset greenhouse gas emissions, remove air and water pollutants, cool the local climate, reduce runoff, and thus, improve well-being [26] (pathway 2, Figure 1). However, most studies have focused only on a single service, and there is a clear lack of studies on ecosystem service trade-offs in urban environments [27,28]. Different ecosystem services may not be provided equally in space, and some may trade-off with other services or sustainable development objectives. For instance, covering a roof with solar panels can produce green energy, but it may not provide other services such as runoff reduction or urban heat island mitigation. The design and dosage of green infrastructures can also influence the quality of services. For instance, although planting trees can reduce the air temperature, noise, and air pollution, some studies have shown that dense canopies can have the opposite effect [29].

Additionally, increasing the extent and quality of nature in cities can enhance the potential for people to experience or interact with the natural world. These interactions can deliver a wide range of recreational, aesthetic, and spiritual benefits (i.e., cultural services) that can be directly linked to people's health and well-being [30,31] (pathway 3, Figure 1). However, not all interactions are positive, and they may differ according to personal, social, and cultural values. Different groups of people derive distinctly different benefits from interacting with nature and thus seek out different forms of nature [32]. Recently Gaston et al. [33] called for a "personalized ecology", stressing the importance of looking at how people interact with nature to improve urban nature policies and management so that benefits are maximized while costs are minimized. However, research in that direction is still scarce.

## 2.2. Trade-Offs and Synergies among the Three Pathways

The above-mentioned pathways illustrate three different ways of looking for integrating nature in cities, with shifting emphasis on different components of human–nature interactions. If we are to achieve resilient cities, we need to understand how to balance those pathways among themselves, and between environmental and economic development [34]. This is a challenging objective, since there are many synergies [13,14], but also some trade-offs between the three pathways [27]. For instance, conflicts may arise between the maximization of biodiversity conservation (pathway 1) and individual nature experiences (pathway 3). Pioneering studies demonstrated that land sparing (compact urban development) performs better for regional biodiversity conservation than land sharing (reduced urbanization intensity) [25,35]. Land sharing, however, can enhance biodiversity locally and facilitate people's access to green spaces and, in turn, affects their health and well-being [25,36]. Yet, studies exploring the relationships between species diversity and health and well-being have revealed inconsistent and complex relationships that vary with the taxonomic group, social, cultural and urban context, and other moderating variables [17,37,38]. These relationship can take several forms, and benefits may even decline at a higher density of natural elements [37,39,40]. For instance, a recent study showed that the relationship between stress recovery and plant species richness follows a quadratic function, whereby relaxation increases with plant species richness, is highest at intermediate levels of species richness (32 species), and then decreases [41]. In some studies, well-being was found to be positively associated with the species richness, and in others, it was only related to the species richness perceived by the greenspace users, and was not related to the sampled richness [42–44].

Experiences of nature also have a prominent importance in the construct of an individual's sense of connection to nature and care for the natural world, and this is particularly true of childhood experiences [45]. Implementing some NbS, such as adding green spaces, can encourage people to spend more time outside, enhance emotional connections to nature, and potentially increase people's willingness to protect the natural world. However, evidence shows that simply spending more time outdoors may not be enough to promote a sense of commitment to nature conservation, and there is

a need to provide opportunities for meaningful interactions with nature to develop a willingness to protect nature [46]. Although such interactions may deliver well-being benefits, they may also have negative impacts on biodiversity (e.g., hunting, fishing) [39]. For instance, meaningful experiences with charismatic species may relate people to nature and enhance their well-being, as is apparent in the case in the "Gazelle Valley" in Jerusalem, Israel (Figure 2a). However, such interactions may alter the natural behaviors of species, reducing the effectiveness of protecting wild animal populations in cities. Similarly, while garden bird feeding is beneficial for establishing personal connection with nature (Figure 2b) [47], it can also have detrimental effects on bird population dynamics [48]. Identifying NbS that enhance well-being and emotional connection to nature but have a moderate impact on species thus appears to be challenging and requires further attention. For instance, designing green walls that blossom in spring could offer opportunities for people to smell flowers, and nest boxes for birds could help to attract birds, allowing people to observe them.

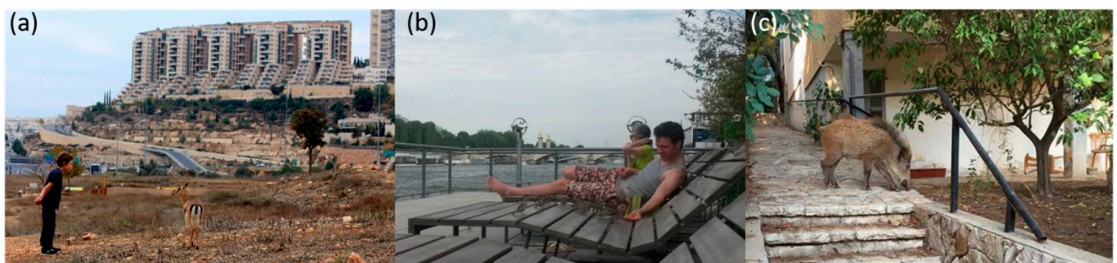

**Figure 2.** Pictures illustrating the different benefits but also trade-offs between and within the different pathways: (**a**) protecting the mountain gazelle in an abandoned agriculture area within the city of Jerusalem, Israel ("Gazelle Valley") can serve as a meaningful experience of nature, but also encourages the imprinting of this endangered mammal; (**b**) bird feeding in Paris, France can help people to establish connections with nature, but may be detrimental for conservation by favoring the prevalence of invasive and generalist species; (**c**) wild boars can elicit people's interest but also cause human–wildlife conflicts (Haifa, Israel).

Additionally, NbS that enhance biodiversity, such as restoring natural vegetation, can favor and even facilitate the spread of species that cause economic, social, and ecological damage (e.g., invasive species and pest species) [49]. Colorful and non-native species in gardens are largely preferred by the public over other native species [50] and have the potential to increase people's sense of connection to nature. However, it remains unknown whether these interactions with non-native species can, in turn, influence people's willingness to protect native species or just increase their preference for non-native ones. Furthermore, public preferences for charismatic and colorful species can facilitate the spread of invasive species, the subset of non-native species that causes ecological damage [51], thereby potentially leading to biotic homogenization [52]. For instance, while there is mounting empirical evidence of deleterious impacts of several invasive parakeets [53], perceptions of those species are conflicting between social groups (e.g., farmers who suffer from the damage caused by parakeets versus park visitors who enjoy interacting with the parakeets) [54]. Using NbS to bring biodiversity back into cities can also result in other human–wildlife conflicts arising from enhanced biodiversity. In Haifa (Israel), for instance, populations of wild boars are growing rapidly, and this has resulted in a human–wildlife conflict that has put the management of natural remnants within the city border into question (Figure 2c) [55]. Similarly, large hyena populations in Ethiopia concentrate around urban areas and cause human–wildlife conflicts [56]. Reintroduction of a native parrot in Wellington city (New Zealand) initiated a feeding-induced wildlife–human conflict: parrots induced property damage, and bird feeding increased the likelihood of this type of degradation [57].

Designing NbS to provide regulating or provisioning services (pathway 2) can benefit biodiversity (pathway 1) under certain circumstances. For instance, nutrient buffer strips or crop diversification may increase the capacity of cropland for infiltration and thus decrease water runoff, which, in turn,

enhances biodiversity and the habitat for pollinators [13]. However, many conflicts have also been highlighted between those pathways. For instance, urban agriculture leads to provisioning services but uses areas that could be dedicated to recreation or habitat restoration for wildlife. A previous study comparing ecosystem service maps with the global distributions of conventional targets for biodiversity conservation showed that areas selected for high biodiversity did not coincide with areas selected for high levels of ecosystem services [58]. Planting tree parks can help to offset carbon emissions and the urban heat island effect and provide a habitat for native species, yet it may limit recreational opportunities for people who like sunbathing on lawns. Implementing NbS can also have negative consequences, i.e., ecosystem disservices. For instance, planting some species of trees can increase allergens (e.g., pollen), decrease, or even inhibit, human mobility and safety (e.g., falling tree limbs) and host pathogens or pests [59]. Trees can also have mixed effects on individuals with respiratory illness, asthma, or allergies [60,61]. Research on urban tree planting programs has revealed a variety of trade-offs between ecosystem services and priority planting locations [62].

Finally, NbS that promote biodiversity conservation, provisioning, or regulating services may not automatically increase the well-being of residents or influence the way people experience nature and their willingness to protect it [5]. A conflict may also arise between designs of urban forms that promote large-scale or even global ecological or environmental benefits and local-scale ecological and social benefits. Research on the trade-offs and synergies within and between the three pathways remains scarce, despite its importance for urban planning [5,27,63]. Furthermore, NbS that increase the green index of a city or neighborhood may result in undesired green or ecological gentrification (i.e., social inequities in greenspace access) and therefore, may not be socially sustainable [64]. In many large cities, minority communities and socially disadvantaged ones often have lower levels of access to urban green spaces [65]. The presence of homeless people who have chosen to live in urban green spaces because all other options are not viable for them can also increase tensions in those spaces [66]. Paradoxically, the implementation of NbS to address the environmental justice problem can increase a neighborhood's health and esthetical attractivity, in turn, also increasing housing costs and property values, and therefore strengthening ecological gentrification [65]. Cities are built for humans, and most local authorities often value human health, recreational opportunities, and equality issues more than the broader public good of conservation [67]. As highlighted in this section, the alignment of those two agendas is challenging. As cities continue to grow worldwide, land will become a more restricted and expensive resource, and this will strengthen the competition between economic and environmental motivations. Identifying how to implement and manage NbS in a way that maximizes the co-benefits derived from those three pathways is crucial to achieve the formation of sustainable and resilient cities.

## 3. Towards a More Integrated Approach: A Novel Framework

These fundamental knowledge gaps, complex relationships, and trade-offs within and between pathways demonstrate that the relationship between NbS and their co-benefits is more complex than commonly argued. Causal understanding of these pathways as well as their moderating variables is key to the design of cost-effective NbS. Raymond et al. [3] recently highlighted the most important gaps in the existing frameworks. They argued that current approaches are insufficient, as they do not allow the integration of co-benefits or the exploration of trade-offs across spatial contexts [3]. Also, although NbS are being increasingly used across cities, their performance is usually evaluated based on broad targets that are not grounded on a deep understanding of the underlying processes. Thus, they have a considerable risk of falling short of their objectives and of eroding public support in their wake. To overcome this challenge, we suggest a paradigm shift in this field, recognizing that there is no "one-size-fits-all" solution and that nature does not exist solely for our benefit. Various interactive social-ecological frameworks have recently been used to jointly assess ecosystem services and biodiversity while considering socio-economic aspects (e.g., [68–70]). However, those frameworks do not integrate the third pathway, implementing NbS to provide nature experiences, and important aspects of human health and well-being and indirect conservation benefits are omitted. We propose a

novel approach that can direct both research and practice on how integrative implementation of NbS can be achieved, aligning the three pathways, by (1) spatially modeling where species will be under different NbS implementation scenarios; (2) predicting, in a comparable manner, how implementing NbS can influence where people will be healthy and happy. This can be achieved by identifying the functional relationships between NbS and various outcomes using experiments and observational approaches that allow scaling up and generalizing beyond a specific urban or socio-cultural context; and (3) using this information in decision support tools that use optimization algorithms to identify planning scenarios that maximize co-benefits (Figure 3).

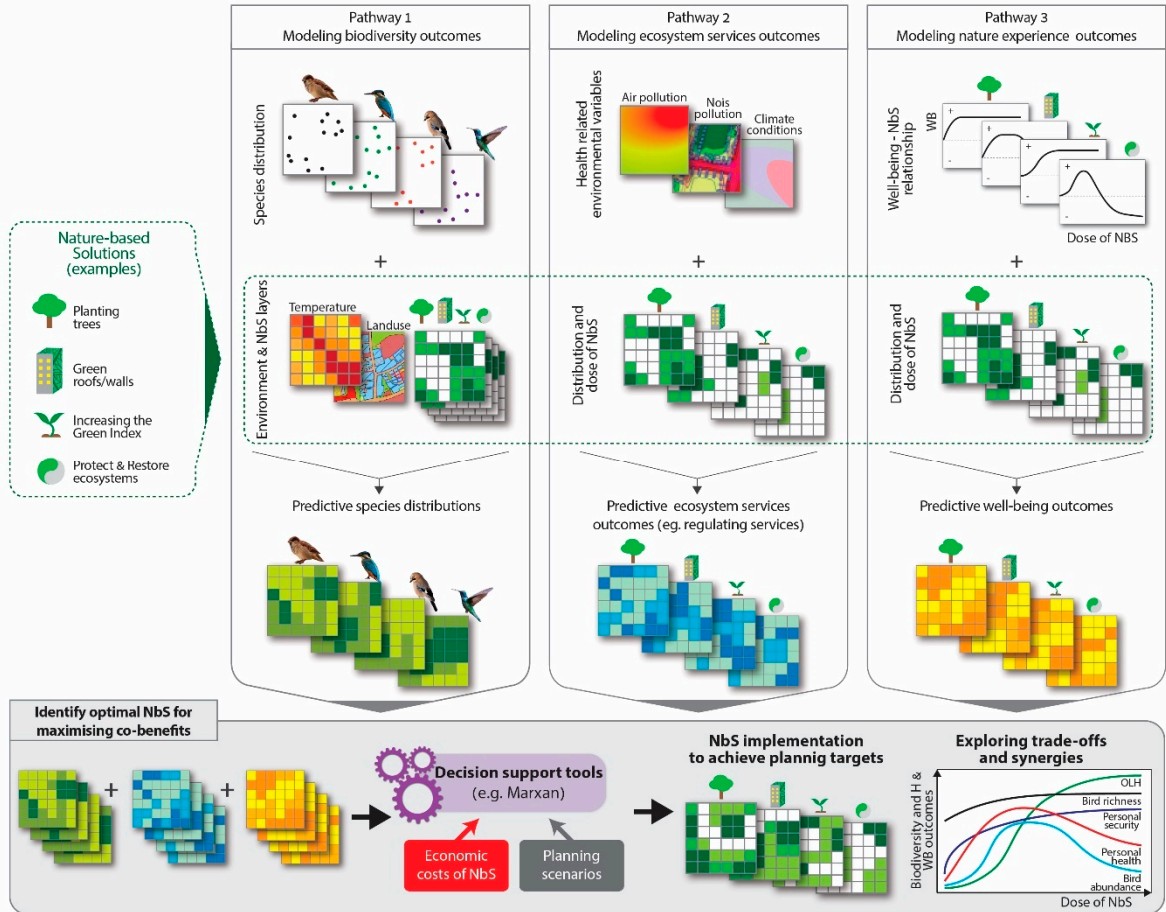

**Figure 3.** Novel integrated approach for (1) modeling biodiversity outcomes (e.g., using species distribution models; pathway 1); (2) modeling ecosystem services outcomes (e.g., using health-related environmental variables; pathway 2); (3) modeling nature experience outcomes (e.g., by designing experiments that identify the functional relationship between NbS and the well-being response, and between NbS and the connection to nature); (4) combining spatial predictions of those outcomes and exploring them against each other to identify trade-offs and synergies between pathways; and (5) using systematic conservation planning tools (e.g., Marxan [71]) to identify the optimal solution for specific city that balances environmental, well-being, and socio-economic considerations.

## 3.1. Predicting Where Species Will Be

Over the last two decades, advances in the development of species distribution models (SDMs) and the availability of fine-scale environmental and ecological data has helped to map species distributions to understand and predict how the implementation of NbS can enhance biodiversity on a city or regional scale (Figure 3, pathway 1). These models combine data on the environmental characteristics of sites in which a species is known to occur to determine the ecological niche of the species. Once the

ecological niche is defined, a predictive map is produced by determining all geographical locations whose environmental characteristics fall within the species' niche [72,73]. Beyond their fundamental role in autecology, SDMs are currently the main tools used to assess the impacts of climate change and land use on biodiversity on a large scale [74]. This shows that they can be used to support conservation decision making by evaluating the consequences of alternative actions (e.g., NbS implementation). SDMs can be used for any species as long as there are sufficient data on their distribution and the relevant environmental predictors [75]. However, despite their numerous applications, to date, only few studies have used fine-grain SDMs within urban areas, and we are aware of few studies that have applied this approach to explore the impacts of different planning scenarios on biodiversity at a city scale (e.g., [25]).

For instance, Sushinsky and colleagues [25] used SDMs to demonstrate the benefits of compact urban development for urban-sensitive bird species compared with sprawling development which increased the distributions of non-native species. SDMs have also been useful at more local scales to demonstrate that dynamic land use changes in business areas (turnover from brownfield to developed sites and vice versa) has resulted in an increase in biodiversity [76,77] or to demonstrate environmental injustice in a local area, with residents of low to mid income tracts being further away from open spaces and thus exposed to less biodiversity and tree canopy [78]. SDMs helped to predict the habitat shifts of two range-expanding native and non-native species [79] and to produce risk maps for two mosquito species that serve as vectors for dengue fever and West Nile Virus [80]. NbS in cities can provide potential habitats acting as inter-population connectivity corridors, and SDMs can be used to derive fine-scale potential connectivity models which allow the identification of connected and completely isolated populations within a study area to detect small linear structures that are important for the inter-population connectivity of specific species [81]. These examples demonstrate the utility of SDMs in mapping and predicting where species will be in the future under different planning scenarios and thus to understand which NbS promote the best biodiversity. Experimental studies empirically testing the functional relationships between NbS and biodiversity can complement modeling efforts and improve our ability to generalize beyond a specific context (Figure 3, pathway 1). The increasing implementation of NbS in cities across the world [7,82], coupled with technological advances that help to produce fine-grain environmental data (e.g., LiDAR technology [83]), offers excellent opportunities for experimental studies that will shed light on the value of NbS for biodiversity conservation.

### 3.2. Predicting Where People Will Be Healthy, Happy, and Connected to Nature

Since cities are built primarily to support humans, it is essential to understand how NbS can affect the geography of health and happiness. Nature is a multifaceted concept that provides detrimental as well as beneficial effects on people's quality of life in different contexts. To date, the benefits of nature have usually been explored through the ecosystem goods and services they provide (Figure 3, pathway 2) and less attention has been paid to their direct impact on people's psychological and physical health (Figure 3, pathway 3). This is because the mapping of ecosystem services with clear physical indicators, such as air pollution, is becoming easier to quantify with satellite and big data [84,85] (Figure 3, pathway 2). Even for regulating services for which information is often not readily available, proxies can be derived from model outputs to quantify them. For instance, air purification services can be assessed using the quantity of air pollutants captured by leaves [86]. In contrast, the direct benefits of nature on people's psychological and physical condition are much more complex and less tangible. Nevertheless, a wide variety of instruments, relying on various psychological scales, is available to measure those concepts. For instance, well-being can be measured using the Positive and Negative Affect Scale [87], the Neighborhood Well-Being Scale [88] or the Warwick Edinburgh Mental Well-Being Scale [89]. The sense of connection to the natural world reviewed by Tam [90] has been assessed through scales on nature relatedness or environmental identity [91,92]. Health can also be assessed with psychological scales, like the Depression, Anxiety, and Stress Scale (e.g., in [93]), or with clinical measures, e.g., blood pressure [94]. Accordingly, mounting empirical evidence demonstrates

that interaction with nature delivers a range of measurable well-being benefits to humans (reviewed by [17,95]).

In contrast with ecosystem services, where the ecosystem functions underpinning the service delivery are often understood, little is known about the nature dosage required to ensure the provision of health and well-being benefits [37]. This is because nature is often explored as a "black box", with studies interchangeably looking at nature and green spaces [17], without identifying which components of nature or ecosystem functions provide those benefits. The recently proposed nature dose-response framework calls for hypothesis-driven science to establish functional relationships between nature, health, and well-being outcomes [37]. Dose-response modeling, which involves modeling people's health responses to a dose of a substance or activity, is a common practice in public health science [96]. Accordingly, both the duration and intensity of activity in nature can improve measures of well-being (self-esteem and mood) [97].

To date, only a few pioneering studies have used this approach to identify the functional relationships between the quantity and, to a lesser extent, the nature intensity and multiple health measures (e.g., [93,98]). These studies, which remain correlative, have demonstrated inconsistent results, especially in relation to benefits derived from the quality of nature (e.g., species diversity) and under different socio-economic and urban contexts [17]. To achieve this, we suggest breaking the generic term "nature" into more practical pieces that can serve as NbS (see Table 1 for examples). In other words, we should move away from simply understanding whether nature benefits urban dwellers to understanding which specific components of nature can provide distinctive social benefits and detriments and how, as well as assessing the quantities in which these should be provided or which level of conflict is tolerable [37] (Figure 3, pathway 3). These components of nature have different ecological values and functions that can change over time, social and urban context; for instance, the ecological value of lawns might increase with age. Previous research has proposed the concept of "service-providing units" to define elements that provide ecosystem services and suggested that the provision of services is likely to differ depending on the scale considered. For instance, a given tree species may provide one type of service if considered on a global scale, but another type of service if considered at a regional scale [99,100].

**Table 1.** Examples of potential specific components of nature, possible ecological indicators, and methods that can be used to measure those indicators.

| Components of Nature | Ecological Indicators | Suggested Method |
|---|---|---|
| Extent of lawn | The extent of lawn per unit area or grid square | The cover of visible lawn can be measured in a combined field and Geographic Information System (GIS) survey using transects or grid cells, depending on the scale and extent of lawn cover [101]. |
| Trees | Density of trees; tree cover and richness; tree cooling effect; tree uptake of air pollutants | Tree cover/density/richness can be estimated using GIS and using the tree layer of the considered area [102]; the tree cooling effect can be estimated using GIS survey using grid cells, depending on the scale and number of trees [103]; the tree uptake of air pollutants can be measured through chemical analyses [104]. |
| Plants and specific flowers | Plant or flower richness, abundance, and evenness; number of rainbow colors; vegetative heterogeneity | Field survey using (1) transects to sample shrubs and visible morphs of flowers; (2) quadrats to sample small herbaceous plants. The standard deviation of Normalized Difference Vegetation Index (NDVI) can be used to estimate the vegetative heterogeneity [93]. |
| Charismatic and colorful animals | Abundance of charismatic and colorful species | Field survey using transects or point counts [105]. |

Adopting a more experimental approach and using big data can help to advance our understanding of the functional relationships between different components of nature or specific intervention that can serve as NbS and well-being outcomes. With the growing recognition of the importance of NbS, many cities in Europe and across the world are currently implementing NbS [82]. This can serve as an excellent "lab" for testing the outcomes of different interventions under various urban and

social contexts. Alternatively, we can use field experiments or virtual reality technology in which the dose of a nature component is manipulated to explore people's responses to a given dosage. Repeating such experiments under different socio-cultural and urban contexts will help to develop a profound understanding of those functional relationships, which is needed to generalize beyond a specific context. We can also build on the idea of "smart cities" (i.e., "wiring the city", partly through information and communications technologies, to improve human and social capital) to design ways of exploring specific components of nature. For instance, linking people's preferences for landscapes to meaningful experiences through analyses of big data of elements on pictures or text contents in social networks (e.g., Flicker, Instagram). Building smartphone applications designed to improve users' experiences in the city could also provide data on the frequency and duration of use of natural settings that can serve as NbS, to identify those that are preferentially used and how they contribute to people's well-being (e.g., "Mappiness" and "iNaturalist" smartphone applications [54,55]). From an applied perspective, this knowledge can help planners to improve the quality of NbS that were originally planned based on a single pathway.

Once the functional relationships between the different doses of NbS or different components of nature and multiple well-being outcomes are established, they can be used to predict the spatial distributions of health and well-being outcomes in relation to nature. This can be achieved by using process-based SDMs that incorporate the current distribution of NbS and the functional relationships into the model [106] to predict the current state of various social outcomes. A more traditional, complementary approach to map social outcomes is to build SDMs which combine real-life observations from a social survey that measures multiple longitudinal health and well-being outcomes across a city with various moderating socio-economic and environmental variables (Figure 3). We can also combine the two approaches by integrating the predictions from the process-based SDMs in the correlative one to form a hybrid model [106,107]. The advantage of this approach is that it allows the integration and validation of the previous findings by exploring the importance of the different predictors that enter the models and their interactions with moderating variables. This can increase the understanding of the significance of the social context on outcomes as well as the importance of the different mechanisms that relate NbS to health and well-being benefits. One of the advantages of using SDMs to predict social outcomes is that it standardizes multiple outcomes of NbS or nature components in a comparable manner.

### 3.3. Identifying Optimal NbS Implementation That Maximizes Co-Benefits

Once the outcomes of the three pathways have been spatially predicted in a comparable manner, we can explore trade-offs and identify synergies between the social and ecological outcomes of NbS and identify optimal planning scenarios that maximize co-benefits (Figure 3). A simple cost–benefit analysis is insufficient for evaluating the cost-effectiveness of NbS for multiple co-benefits, and there is a need for the development of novel tools that consider the spatial distribution of benefits for NbS design, implementation, and acceptance [3]. Such tools already exist in the discipline of Systematic Conservation Planning (SCP), but, to our knowledge, their use in urban contexts has been restricted to species-based approaches [108]. These tools allow the exploration of how to implement NbS to meet a given target under specific budget constraints and maximize co-benefits to achieve the multifunctional objective which is the basis of the NbS concept [1]. We need to develop optimization models or use and adapt existing decision support tools (e.g., Marxan [71]) for which algorithms compare alternative planning scenarios to identify optimal ones that meet defined targets, while minimizing the overall cost (Figure 3). Knowledge about functional relationships between NbS and multiple dimensions of biodiversity health and well-being, together with the spatially explicit maps of species distributions and people's health and well-being, can be used as input in those models or tools. This can be used by planners who seek to integrate NbS in urban planning in a way that optimizes benefits and mitigates societal challenges.

One challenge for sustainability planning is that similar solutions cannot always be applied in different cities, especially when considering changes in a climate or national context. For instance, shading elements are particularly appreciated in hot climates but not in high latitude countries with limited amounts of daylight. In addition to the obvious differences in ecosystem compositions and dynamics, there are also cultural differences. For instance, studies have found that the functional relationship between the preference for landscape and the biodiversity level varies between countries [38] and types of landscape [109]. Some components that are beneficial in one part of the world may not be in another. The framework we propose takes these complexities into account to create tailor-made NbS for a given socio-ecological context. Nonetheless, as soon as empirical data on specific components of nature that can serve as NbS begin to accumulate, we believe that some generalities could emerge. Finally, it is also important to emphasize that the identification of functional relationships does not mean that these relationships are fixed in time or space. Understanding how some specific components of nature serving as NbS are perceived to benefit or harm people is key for promoting policies, conservation outreach, and education interventions aiming to change these relationships. For instance, an education campaign about wild boars in Haifa, Israel (Figure 2c) or about the Andean bear in Ecuador [110] may help reduce human–wildlife conflicts and reshape the functional relationship between the abundance of these species and human well-being. Quantifying the functional relationships will not only highlight conflicts but can help to advance the development of cost-effective actions to mitigate them, as the outcomes of interventions can be measured and be compared to a general or case-specific baseline.

## 4. Discussion and Conclusions

The main paradigm in urban ecology is gradually shifting [111]. The first generation focused on ecology *in* cities and was aimed at understanding ecological processes in different patches within cities and along the urban–rural gradient. Focus on the ecology *of* the city emerged in the late 1990s, where entire urban mosaics were treated as social-ecological systems and the paradigm was changed from a focus on biotic communities to holistic social-ecological systems [112]. We are now witnessing the third generation of urban ecological research—ecology *for* the city [113]—which emerged within the context of urban sustainability and encourages ecologists to engage with other specialists and urban dwellers to shape a more sustainable urban future. In this essay, we adopt the ecology *for* cities paradigm, discuss the challenges, and propose solutions for bringing nature back into cities and using it to mitigate some of the local and global detrimental impacts of urbanization. This may well be one of the greatest challenges for humans ahead [5,34].

NbS is one of the most promising ways to meet this challenge, but as we demonstrated, despite the many synergies between the pathways (Figure 1), there are also a variety of trade-offs that threaten the ability of NbS to simultaneously deliver social, ecological, and environmental benefits. The framework proposed in this essay highlights these complexities, and instead of ignoring them, suggests a way to incorporate them into future research and practices to advance the development and implementation of NbS that maximize co-benefits. Previous social-ecological frameworks were recently proposed to jointly assess ecosystem services and biodiversity, considering the socio-economic constraints and feedback within the system in an interactive process [68–70]. These frameworks are useful, as they integrate spatial modeling and decision support tools in dynamic social-ecological systems. However, they focus on only part of the system, omitting important aspects of human health and well-being as well as potential indirect conservation benefits and therefore cannot provide a holistic picture. The gaps in existing frameworks were recently summarized by Raymond et al. [3], highlighting the importance of assessing the co-benefits and costs of NbS across elements of socio-cultural and social-economic systems, biodiversity, ecosystems, and climate. While the above-mentioned frameworks suggest ways to advance our understanding and implementation of NbS, they do not describe *how* this can be achieved.

Our solution-oriented framework offers a means by which future research and practices can shed light on context-specific needs and identify synergies and trade-offs between different outcomes of NbS to maximize co-benefits. To date, the importance of co-benefits has been well acknowledged in the NbS literature [3,13,14], but as far as we know, existing frameworks do not suggest how we can go beyond assuming co-benefits across domains. The proposed framework is applicable to any NbS outcome, although in this essay, we chose to demonstrate its applicability to human–nature interactions (pathway 3). This is because we agree with others who believe that it is crucial to place people at the center to design sustainable cities [3,37]. The only way to achieve this is by integrating methods from several disciplines, as our framework proposes. First, the niche theory from ecology and the cutting-edge tools it provides allow various outcomes (positive and negative) of existing and potential NbS to be mapped in a comparable manner. Second, experimental and observational studies that adopt the dose–response approach (from public health sciences) can help to identify the functional relationships between NbS and health and well-being outcomes across social and urban contexts. Accumulating sufficient knowledge of those relationships will allow generalization beyond a specific context. These generalizations can only take us part of the way toward sustainability, as we argue that there is no one-size-fits-all solution, and NbS need be tailored to fit a specific context by prioritizing local targets and budget constraints. This is where decision support tools (e.g., [71]) that use optimization can come in handy to identify which, where, and how many NbS should be implemented to meet given targets and maximize co-benefits.

To conclude, cities are centers of population and education, and the way that they will look in the future will influence the direction our world will take. We hope this essay will advance the knowledge of the ecology *for* cities paradigm and pave the way toward bringing nature back into cities in a way that benefits people and biodiversity conservation by (1) encouraging interdisciplinary research that seeks to understand the mechanisms driving the relationships between NbS and various outcomes; (2) helping researchers and planners to recognize the importance of understanding and taking into account trade-offs and synergies between pathways to inform decisions. This can help to identify local "win-win" or "lose-lose" situations, as well as "low/high hanging fruits", i.e., opportunities with small costs but small benefits, or in contrast, opportunities that are very expensive in terms of NbS implementation but with large benefits; and (3) encouraging planners to set up targets and identify NbS that maximize co-benefits rather than assuming them.

**Author Contributions:** Conceptualization, A.C. and A.S.; writing—original draft preparation, A.C. and A.S.; writing—review and editing, A.C. and A.S.; supervision, A.S.; funding acquisition, A.S. and A.C.

**Funding:** This research was funded by the Israel Science Foundation (Grant No. 1456/16) and A.C. was also partly supported by the Technion Fellowship for distinguished postdoctoral fellows.

**Acknowledgments:** We wish to thank A. Turbe, M. Jeanmougin, Y. Depietri, R. Cohen and D. Orenstein for their insightful comments on earlier versions of the manuscript.

**Conflicts of Interest:** The authors declare no conflict of interest. The funders had no role in the design of the study, in the writing of the manuscript, or in the decision to publish this conceptual framework.

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
