# Peer review of "Beyond Assuming Co-Benefits in Nature-Based Solutions: A Human-Centered Approach to Optimize Social and Ecological Outcomes for Advancing Sustainable Urban Planning"

_sustainability, doi:10.3390/su11184924_

Round 1

Reviewer 1 Report

The manuscript presents a new framework to examine how nature-based solutions and urban greening can be implemented in cities.  Although I can see how the framework can be useful – especially figure 1 paired with figure 3 – I think much of the paper needs to more clarity in the description of the framework as well as a greater discussion of the tradeoffs to show how the complexity can be addressed.

Below I list a number of changes that could help increase the specificity and level of context of the text:

Lin 63-65 – need citations to support and show the documentation of NBS for these cities

Line 70, Figure 1 - I think the discussion of why this particular framework, who this framework is for, and how this framework would be used needs to be clarified.  For example, the authors describe three pathways that represent motivations for urban greening.  From the figure, I’m guessing that these are motivations by urban planners, but could they be motivations by urban citizens as well (think private land, community gardens, etc.). Are these motivations well documented in the literature and how are they currently being enacted?

Additionally, are the pathways entirely distinct? In the ecological literature, there is quite a bit of overlap between biodiversity and ecosystem provision as well as biodiversity and nature relationships. It would be interesting to discuss the complexities of these overlaps as well as why the authors believe they should be put into distinct pathways in this framework.

This leads to questions in figure 1 – the top box should say urban planners? And maybe you could lable the second row as “motivations for urban greening or nature based solutions”. IT seems like the last box should be “improved environmental and social health or outcomes”.

Figure 3 is quite clear and useful and it is helpful toward the overall understanding of the framework and how someone could use it.  I’m wondering if this should be brought forward and made more prominent as a the central focus of the paper.

Figure 3 – An explanation of Marxan should be in the figure caption

Line 88 - This needs to be unpacked. Give examples on where assumptions have fallen over or where pathways did not develop as planned. 

Otherwise - I suggest that a larger section devoted to tradeoffs and going into more detail about each of the tradeoffs and how they have been addressed in the past is needed.  You could bring in some of the literature that you have in the discussion as well regarding aspects such as human-wildlife conflict.  You have quite a developed section on spatial issues – such as the different examples of land sharing versus land sparing.  Having a equal discussion on human-wildlife conflict would be really interesting.  Invasive species could be part of this discussion.  I remember an article on clearing of rats from Wellington as a way to bring back more native wildlife for the nature wildlife connection and needing to create different types of greenspaces that cater to native wildlife.

Line 460 - There are examples of using deciduous trees to help with these trade-offs - as well as research that takes into account the different ways of planting around the house to take advantage of sun and shading. Perhaps some of this could be considered in the tradeoff discussion.

Line 462 - Provide some examples of how cultural difference may warrant different solutions.

Author Response

The manuscript presents a new framework to examine how nature-based solutions and urban greening can be implemented in cities.  Although I can see how the framework can be useful – especially figure 1 paired with figure 3 – I think much of the paper needs to more clarity in the description of the framework as well as a greater discussion of the tradeoffs to show how the complexity can be addressed.

We clarified the description of the framework and provided more discussion on the trade-offs and how our novel framework allows to integrate the trade-offs to better understand and address complexities. In short, we highlighted that previous research by Raymond and colleagues [1] highlighted what needs to be done and listed three major gaps: (1) consider different social and environmental contexts, (2) transfer and upscale NbS, and (3) integrate optimization tools to inform decision making. Here, we suggest how this can be achieved, and propose a novel framework to bridge these three major gaps (lines 86-97 in revised manuscript). This new framework is useful for both scientists and practitioners: it can direct empirical research on functional relationships between NbS and social and ecological outcomes and provide guidelines for planners on how to implement NbS that jointly provide social and ecological benefits (lines 97-102 in revised manuscript).

Below I list a number of changes that could help increase the specificity and level of context of the text:

Line 63-65 – need citations to support and show the documentation of NBS for these cities

According to the reviewer suggestion, we added a reference to support this statement (line 46 in revised manuscript).

Line 70, Figure 1 - I think the discussion of why this particular framework, who this framework is for, and how this framework would be used needs to be clarified.  For example, the authors describe three pathways that represent motivations for urban greening.  From the figure, I’m guessing that these are motivations by urban planners, but could they be motivations by urban citizens as well (think private land, community gardens, etc.). Are these motivations well documented in the literature and how are they currently being enacted?

We clarified these aspects in the revised manuscript. Motivations for urban greening can be from planners, policy makers or researchers, but also the general public. Our novel framework is useful for both scientists and practitioners. One the one hand, it suggests to direct empirical research on functional relationships between NbS and social and ecological outcomes and on the other it offers to use decisions support tools to guide planners on how to implement NbS that jointly provide social and ecological benefits (lines 97-102 in revised manuscript). We clarified the figure to illustrate the three main motivations. We specified this and provided references supporting this in the revised manuscript (lines 51-53), but since we shortened this section (as suggested), this is not developed further in the manuscript. 

Additionally, are the pathways entirely distinct? In the ecological literature, there is quite a bit of overlap between biodiversity and ecosystem provision as well as biodiversity and nature relationships. It would be interesting to discuss the complexities of these overlaps as well as why the authors believe they should be put into distinct pathways in this framework.

We thank the reviewer for this comment, which pushed us to better formulate our ideas. The three pathways are not entirely distinct and of course there are overlaps, synergies and trade-offs. However, we think it is more straightforward to first explain the key pathways as distinct and then present the complexities and overlaps. Following the reviewer comment we clarified in the introduction section of revised manuscript that the three pathways are not entirely distinct, but present three different ways of looking at integration of nature in cities, with shifting emphasis on different components of human-nature interaction (lines 63-66). We also briefly explain that we discuss these synergies and tradeoffs within and between the pathways in the in the “synthesis of the literature: pathways, synergies and tradeoffs” section (line 100).

This leads to questions in figure 1 – the top box should say urban planners? And maybe you could label the second row as “motivations for urban greening or nature based solutions”. IT seems like the last box should be “improved environmental and social health or outcomes”.

We edited the figure to clarify these aspects.

Figure 3 is quite clear and useful and it is helpful toward the overall understanding of the framework and how someone could use it.  I’m wondering if this should be brought forward and made more prominent as a the central focus of the paper.

Following the reviewer suggestion, we gave more importance to this figure which represents our novel framework and added more references to figure 3 in the text. In the revisions we also shorten the sections about the pathways to ensure that our framework is not hidden by long literature synthesis given other comments of both reviewers.

Figure 3 – An explanation of Marxan should be in the figure caption

We changed the icon related to Marxan in the figure, for a more general icon on decision support tools, and mentioned Marxan as an example in the figure caption.

Line 88 - This needs to be unpacked. Give examples on where assumptions have fallen over or where pathways did not develop as planned. 

Here (line 75 in revised manuscript), the research by Dearborn and Kark [2] that we cited (number 10 in revised manuscript) provides an example that advocate for win-win solutions between pathways. We provided additional citations of studies that already suggested this ideas in the following sentence [3,4]. However, the bulk of examples are given in the tradeoffs section later on in the manuscript, and therefore do not want to elaborate more on this issue at this point of the manuscript.

Otherwise - I suggest that a larger section devoted to tradeoffs and going into more detail about each of the tradeoffs and how they have been addressed in the past is needed.  You could bring in some of the literature that you have in the discussion as well regarding aspects such as human-wildlife conflict.  You have quite a developed section on spatial issues – such as the different examples of land sharing versus land sparing.  Having a equal discussion on human-wildlife conflict would be really interesting.  Invasive species could be part of this discussion.  I remember an article on clearing of rats from Wellington as a way to bring back more native wildlife for the nature wildlife connection and needing to create different types of greenspaces that cater to native wildlife.

We reduced sections describing the pathways to develop more the tradeoffs within and between pathways. Particularly, we increased discussion on human-wildlife conflicts, including invasive species, in the revised manuscript (lines 184-202). We raised the issue of public preferences for colorful and non-native species, that can facilitate the spread of invasive species, affecting in turn native populations. We also provided two other examples of human-wildlife conflicts: large populations of hyenas that concentrate around urban areas in Ethiopia, and reintroduction of a native parrot in Wellington City (New Zealand) that induced property damages and thus initiated human-wildlife conflicts.  

Line 460 - There are examples of using deciduous trees to help with these trade-offs - as well as research that takes into account the different ways of planting around the house to take advantage of sun and shading. Perhaps some of this could be considered in the tradeoff discussion.

We developed the trees examples in the tradeoffs section as suggested (lines 211-219 in revised manuscript).

Line 462 - Provide some examples of how cultural difference may warrant different solutions.

We provide two examples of cultural differences in the revised manuscript (lines 432-434), based on preferences for biodiversity in different settings between cultures.

Reviewer 2 Report

I like the topic and the approach proposed by the paper. However, the main issue is that in the construct of the argument, the paper attacks a “strawman”—lack of socio-ecological system models to support nature-based solutions. In fact, I have come across a number of literatures on supporting NBS (such as ecosystem service protection) through the method of social-ecological modeling, for example:

https://www.sciencedirect.com/science/article/pii/S1470160X19301682

https://www.sciencedirect.com/science/article/pii/S1462901117306317

Instead of arguing “the lack of”, I suggest the paper to frame on how the authors’ social-ecological approaches have improved from the above-literatures and beyond as a methodological contribution to existing body of works.

Other detailed comments:

Line 16: It is not ideal to use weak and ambiguous word (might) this early abstract. The authors should use more definitive construct such as “has the potential to… under xxx conditions”.

Line 34: NBS is already a quite established field, thus I do not see the necessity of introducing the topic from such huge background. I suggest directly delving into the NBs discussion and definition from the first paragraph.

Figure 1: Social-ecological model and NBs emphasize the feedback and interactions between the systems; however these are devoid in this graph.

Section 2-3: I think these are better incorporating in a “literature study” or “related works” section. Currently it is hard to distinguish whether these are the authors’ original ideas or a review of the existing literatures—I believe it is closer to the later.

Line 293: Again, if the authors’ review the works of existing social-ecological models in support of NBs, models with interactive systems exist.  

Table 1: I believe most of the works and methods have already been explored and synthesized by the authors, thus the authors should use references to related works in the table.

Line 453: Suggesting the algorithms and models are more important than any specific software, as the academia is moving towards open-source models for social-ecological modeling.

Line 478: This paper lacks a discussion part that compares how this works improve from existing social-ecological modeling efforts and highlight some of the main application fields and policy suggestions from the paper (they do exist somewhere in the paper, but usually readers would like to find these in a discussion sections not anywhere in the paper). For example, previous social-ecological models in urban systems or support urban planning including:

https://link.springer.com/article/10.1007/s11252-016-0574-9

https://www.sciencedirect.com/science/article/pii/S0959652619318104

https://www.sciencedirect.com/science/article/pii/S221067071730094X

Author Response

The revised manuscript was sent for English editing (ref. English-11424).

I like the topic and the approach proposed by the paper. However, the main issue is that in the construct of the argument, the paper attacks a “strawman”—lack of socio-ecological system models to support nature-based solutions. In fact, I have come across a number of literatures on supporting NBS (such as ecosystem service protection) through the method of social-ecological modeling, for example:

https://www.sciencedirect.com/science/article/pii/S1470160X19301682

https://www.sciencedirect.com/science/article/pii/S1462901117306317

Instead of arguing “the lack of”, I suggest the paper to frame on how the authors’ social-ecological approaches have improved from the above-literatures and beyond as a methodological contribution to existing body of works.

We thank the reviewer for this useful comment. In the revised manuscript, we better acknowledged the overlaps between pathways and highlighted the multiplicity of tradeoffs within and between pathways. We emphasized the lack of framework comprehensively considering those three pathways and how these trade-offs can be explored and used to inform decision making. Previous research by Raymond and colleagues [1] highlighted what needs to be done and listed major gaps. Here, we suggest how this can be achieved, and we believe our framework can help bridge these specific knowledge gaps. We clarified this in both introduction and discussion of the revised manuscript. Following the reviewer’s comment we further emphasized in the existence of previous interactive socio-ecological frameworks that allow to jointly explore ecosystem services and biodiversity, while considering socio-economic constraints, but highlighted the absence of inclusion of pathway 3 (providing nature experiences) in those frameworks (lines 465-475 in revised manuscript). Finally, we also dedicated part of the discussion to compare our framework with previous socio-ecological frameworks and explain how our framework is advancing both science and practice (lines 448-505 in revised manuscript).

Other detailed comments:

Line 16: It is not ideal to use weak and ambiguous word (might) this early abstract. The authors should use more definitive construct such as “has the potential to… under xxx conditions”.

We edited this sentence.

Line 34: NBS is already a quite established field, thus I do not see the necessity of introducing the topic from such huge background. I suggest directly delving into the NBs discussion and definition from the first paragraph.

We deleted the first paragraph to directly delve into NbS definition and discussion.

Figure 1: Social-ecological model and NBs emphasize the feedback and interactions between the systems; however these are devoid in this graph.

We edited the figure to demonstrate the interaction between the pathways and feedback loop from outcomes to targets for urban planning.

Section 2-3: I think these are better incorporating in a “literature study” or “related works” section. Currently it is hard to distinguish whether these are the authors’ original ideas or a review of the existing literatures—I believe it is closer to the later.

We merged sections 2 (pathways) and 3 (tradeoffs) into a single section entitled “Synthesis of the literature: pathways, synergies and tradeoffs” and shortened in considerably so the bulk of the text focuses on our novel framework and how it advances science and practice.

Line 293: Again, if the authors’ review the works of existing social-ecological models in support of NBs, models with interactive systems exist.  

We clarified the gaps in existing frameworks and how our approach can bridge those gaps, lines 449-505 in revised manuscript.

Table 1: I believe most of the works and methods have already been explored and synthesized by the authors, thus the authors should use references to related works in the table.

We added references in the table, as examples.

Line 453: Suggesting the algorithms and models are more important than any specific software, as the academia is moving towards open-source models for social-ecological modeling.

We edited this part, highlighted the need to create algorithms and models, or improve existing ones, citing the example of Marxan, and removed specific sentence describing Marxan (lines 418-423 in revised manuscript).

Line 478: This paper lacks a discussion part that compares how this works improve from existing social-ecological modeling efforts and highlight some of the main application fields and policy suggestions from the paper (they do exist somewhere in the paper, but usually readers would like to find these in a discussion sections not anywhere in the paper). For example, previous social-ecological models in urban systems or support urban planning including:

https://link.springer.com/article/10.1007/s11252-016-0574-9

https://www.sciencedirect.com/science/article/pii/S0959652619318104

https://www.sciencedirect.com/science/article/pii/S221067071730094X

Following this very useful comment we changed the ‘conclusion’ section into a ‘discussion and conclusion’ one and discussed how the new framework improves from existing socio-ecological frameworks (lines 448-505 in revised manuscript). We acknowledge the usefulness of these existing social-ecological frameworks but argue that the third pathway is not comprehensively integrated in those frameworks, therefore omitting important aspects of human health and well-being, as well as indirect ecological benefits.

Raymond, C.M.; Frantzeskaki, N.; Kabisch, N.; Berry, P.; Breil, M.; Nita, M.R.; Geneletti, D.; Calfapietra, C. A framework for assessing and implementing the co-benefits of nature-based solutions in urban areas. Environ. Sci. Policy 2017, 77, 15–24.

Round 2

Reviewer 2 Report

The revision has addressed reviewer concerns and can be published.

Author Response

Thank you very much.